# Electrochemical Cytosensor Based on a Gold Nanostar-Decorated Graphene Oxide Platform for Gastric Cancer Cell Detection

**DOI:** 10.3390/s22072783

**Published:** 2022-04-05

**Authors:** Amin Zhang, Qianwen Liu, Zhicheng Huang, Qian Zhang, Ruhao Wang, Daxiang Cui

**Affiliations:** 1Institute of Nano Biomedicine and Engineering, Shanghai Engineering Research Center for Intelligent Diagnosis and Treatment Instrument, Department of Instrument Science and Engineering, School of Electronic Information and Electrical Engineering, Shanghai Jiao Tong University, 800 Dongchuan RD, Shanghai 200240, China; zhangamin@sjtu.edu.cn (A.Z.); liuqianwen@sjtu.edu.cn (Q.L.); orange-.-hung@alumni.sjtu.edu.cn (Z.H.); qianzhang0130@sjtu.edu.cn (Q.Z.); wrh1120@sjtu.edu.cn (R.W.); 2National Engineering Research Center for Nanotechnology, 28 Jiangchuan Easternroad, Shanghai 200241, China

**Keywords:** GO-AuNSs@rBSA-FA, electrochemical cytosensor, DPV, cancer cell analysis

## Abstract

Effectively capturing and sensitively detecting cancer cells are critical to clinical diagnosis and cancer therapy. In this work, we prepared gold nanostar-decorated graphene oxide (GO-AuNSs) nanocomposites using a ultraviolet (UV)-induced strategy, and then modified them with a layer of bio-complex rBSA-FA (coupled reduced bovine serum albumin with folic acid) to generate GO-AuNSs@rBSA-FA nanocomposites. Herein, the application of GO and AuNSs not only strengthened the conductivity of the sensing platform but also guaranteed nanocomposites with biocompatible performance. Moreover, the adopted rBSA-FA layer could effectively enhance the stability and specificity towards gastric cancer cells (MGC-803). According to a systemic construction procedure, a novel electrochemical cytosensor based on GO-AuNSs@rBSA-FA was fabricated for MGC-803 cell detection. With the assistance of cyclic voltammetry (CV) and differential pulse voltammetry (DPV), the cytosensor reached a detection limit of 100 cell/mL in a wide linear range of 3 × 10^2^~7 × 10^6^ cell/mL towards MGC-803 cells. The good electrochemical characteristics for the cancer cell analysis indicate a promising prospect of this electrochemical cytosensor in clinical cancer diagnosis.

## 1. Introduction

The high incidence and fatality rate of cancer make it a major disease that seriously endangers human health [1,2]. Effective capture and sensitive detection of cancer cells are significant for clinical cancer diagnosis and treatment [3]. Up to now, numerous strategies have been developed for cancer cell determination, such as surface-enhanced Raman scattering (SERS), immuno-fluorescent microscopy, fluorescent spectroscopy, and so forth [4,5]. However, the wide application of these methods is sometimes restricted by expensive instruments, complicated pretreatment, strict experimental conditions, and long operation times [6]. In the past few years, cancer cell determination based on electrochemical assay received considerable attention owing to its easy operation, good reliability, high sensitivity, excellent specificity, and satisfactory reproducibility [7]. Hence, excessive efforts have been devoted to develop electrochemical aptasensors, immunosensors, nucleic acid biosensors, and cytosensors [8,9]. Furthermore, the design of folic acid (FA)-modified nanomaterials for cancer cell detection has aroused extensive attention in recent years [10,11].

Numerous nanomaterials such as thiourea-capped MoS_2_ nanosheets [12,13], CuO/WO_3_ nanoparticle (NP)-decorated graphene oxide nanosheets [14], DNA-templated silver nanoclusters [13], and CuO NPs [15] have been reported to capture cancer cells or amplify electrochemical signals during electrochemical detection of cancer cells. Gold nanostars (AuNSs) have gained intensive attention in diagnosis, bioimaging, and biomedical applications [16,17,18]. AuNSs with controllable sizes tend to exhibit easy surface modification, favorable optical properties, and high biocompatible performance [19]. As a novel two-dimensional nanomaterial, graphene oxide (GO) has become a promising candidate in constructing biosensors and carrying metal nanomaterials owing to its large surface area, abundant functional groups, excellent conductivity, high hydrophilicity as well as outstanding mechanical stiffness [20,21]. Studies have demonstrated that employing GO to modify electrode surfaces could effectively optimize the electrical response signal of chloramphenicol [22], ractopamine, clenbuterol [23], hydrogen peroxide [24], hydrazine [25], and so forth [26]. Diverse compatible materials (e.g., ZnO [27], TiO_2_ [28], CuO [29], Co_2_SiO_4_ nanobelts [30]) have been exploited to pair with GO in the fabrication of electrochemical sensors to make full use of their outstanding properties. However, the integration of GO and AuNSs for biological detection is still lacking.

FA is a water-soluble vitamin essential for cellular division and prenatal development [31,32,33]. It can bind to folate receptors (FRs) with a high affinity and then enter the cell by FR-mediated endocytosis [34]. The expression of FRs is limited in normal tissues. However, the overexpression of FRs has been demonstrated in tumor cells, including ovarian, lung, cervical carcinoma, gastric, prostate cancer, and so on [7,35]. The difference in FR expression, along with the high affinity between FRs and FA, has broadened the way towards FA-related targeting therapeutics and cancer cell analysis. For instance, Du et.al designed an electrochemical impedance biosensor based on FA-functionalized zirconium MOFs (FA@UiO-66) for HeLa cancer cell detection [3]. In their work, the detection limit was 90 cells mL^−1^ with a good selectivity. Li et.al found that the capture efficiency of FA-GAM-OA (FA and octadecylamine (OA)-functionalized grapheme aerogel microspheres) was much higher than that of GAM-OA towards HepG2 cell capture [36].

Through a systemic preparation process, we first synthesized gold seeds on the surface of GO (GO-Au) in-situ without adding any traditionally used reductant, surfactant or polymer stabilizers by a UV-induced strategy. Then, according to the commonly seed-mediated method, we prepared gold nanostars (AuNSs) supported on graphene oxide (GO) to generate a GO-AuNS nanocomplex based on the above-acquired GO-Au seeds. In this process, GO played numerous roles as a surfactant to protect Au seeds as well as Austar from aggregation and as a reductant to synthesize Au seeds as well as carriers to support biomolecules. Based on the amidation reaction between the amino group of reduced bovine serum albumin (rBSA) and the carboxyl group of FA, rBSA-FA was obtained. The physical adsorption of GO and synergy with the Au-S chemical bond between AuNSs and rBSA promoted the combination of GO-AuNSs and rBSA-FA to produce GO-AuNSs@rBSA-FA nanocomposites, which were employed to modify the working electrode for cancer cell detection. Hence, we designed a new type of electrochemical sensor using the three-electrode system for gastric cancer cell detection basing on GO-AuNSs@rBSA-FA nanostructures (Figure 1). In this electrochemical system, GO not only increased the specific surface area but also enhanced electrochemical conductivity with the synergy of AuNSs. In addition, rBSA effectively improved the biocompatibility and stability of the GO-AuNSs. The introduction of FA molecules enhanced the specificity of the nanocomposites. Finally, the cytosensor in this proposal allowed quantitative detection of cells in a wide linear range of 3 × 10^2^~7 × 10^6^ cell/mL with a limit of 1 × 10^2^ cell/mL (S/N = 3), showing good electrochemical properties for MGC803 cell detection.

## 2. Experimental

### 2.1. Materials and Reagents

Bovine serum albumin (BSA) was obtained from Aladdin Reagent Co., Ltd. (Shanghai, China). Chloroauric acid (HAuCl_4_·3H_2_O) was purchased from Sigma–Aldrich Trading Co., Ltd. (Shanghai, China). Graphene, sodium borohydride (NaBH_4_), dimethyl sulfoxide (DMSO), hydrochloric acid (HCl), potassium dihydrogen phosphate (KH_2_PO_4_, 0.1 M), disodium hydrogen phosphate (Na_2_HPO_4_, 0.1 M), and folic acid (FA) were all purchased from Sinopharm Chemical Reagent Co., Ltd. (Shanghai, China). N-hydroxysuccinimide (NHS), 1-(3-dimethylaminopropyl)-3-ethylcarbodiimide hydrochloride (EDC), pancreatin, penicillin, and streptomycin, high sugar medium, and fetal bovine serum were all obtained from Thermo Fisher Scientific Co., Ltd. (Shanghai, China). A phosphate-buffered saline (PBS, 100 mM) solution was prepared by mixing Na_2_HPO_4_ and KH_2_PO_4_ in a ratio of 1:1. Deionized water was prepared using a Millipore economical ultra-pure water machine (Massachusetts, USA) with an electrical conductivity not lower than 18.2 MΩ. 

### 2.2. Apparatus

Scanning electron microscopy (SEM) measurements were performed on a JEOL JSM-7800F Prime. Energy dispersive spectrometer (EDS) information of the nanocomposites was obtained by a Thermo Scientific NORANTM System 7 EDS. UV-vis absorption spectra were obtained from a VARIAN Cary 50 Conc UV-Visible Spectrophotometer. Transmission electron microscopy (TEM) data were obtained by using a TALOS F200X. Electrochemical measurements were conducted on a CHI 660D electrochemical workstation from Shanghai Chenhua Apparatus Inc., Shanghai, China. 

### 2.3. Preparation of GO-AuNSs@rBSA-FA Nanocomposites

#### 2.3.1. Preparation of GO-AuNS Nanocomposites

The GO-AuNS nanocomposites were prepared according to the seed-growth method in which Au nanoparticles modified GO as the seeds. At first, the GO was synthesized according to a previously reported method [37]. Briefly, potassium permanganate and concentrated sulfuric acid oxidized the flake graphite. Then, solid GO dispersed in water was obtained by ultrasound. After that, to obtain GO-Au NPs acting as seeds, the GO solution (0.25 mg/mL, 20 mL) was mixed with HAuCl_4_·3H_2_O (242.81 mM, 66 μL); the pH of the mixture was adjusted to 11 using NaOH (1 M). Finally, GO-Au Seeds were obtained after the mixture was irradiated in a molecular hybridization instrument with a 1000 J UV lamp for 2 h. The GO-Au seed solution was stored in the dark for further use.

The GO-AuNS nanocomposites were compounded via seed-mediated growth. In brief, 2 mL of the GO-Au seed solution was mixed with 38 mL of deionized water when the magneton churned at a high speed. Then, HAuCl_4_·3H_2_O (242.81 mM, 21 μL), HCL (1 M, 500 μL) and AgNO_3_ (3 mM, 2.2 mL) were quickly added to the mixture. Finally, ascorbic acid (AA, 0.1 M, 1.6 mL) was blended into the solution to synthesize GO-AuNSs. Finally, the solution was centrifuged and washed with deionized water to obtain GO-AuNSs, which were then dissolved in PBS solution (pH = 7.4) and stored in the dark for future needs.

#### 2.3.2. Preparation of rBSA-FA Nanocomposites

The rBSA-FA nanocomposites were prepared according to previous work with minor revision [38]. Firstly, 1 g of FA powder that dissolved in the DMSO solvent was mixed with EDC and NHS in a ratio of 1:2.5:2.5. The mixed solution was shaken for 12 h in a dark environment at room temperature. Then, acetone and ether were mixed in a volume ratio of 3:7 for centrifugation and washing of the above products. After the activation treatment, FA-NHS powder was prepared by freeze-drying activated FA. Following that, NaBH_4_ (1 M, 250 μL) was dissolved in BSA (4 mg/mL, 10 mL) to obtain rBSA solution. Two hours later, 50 mg of FA-NHS powder was added to 10 mL of rBSA and coupled for 8 h. After excess FA-NHS was purified by ultrafiltration, the rBSA-FA solution was finally obtained. The freeze-dried rBSA-FA powder was stored in a 4 °C refrigerator for future needs.

#### 2.3.3. Preparation of GO-AuNSs@rBSA-FA Nanocomposites

The GO-AuNSs@rBSA-FA nanocomposites were synthesized by connecting rBSA-FA with GO-AuNSs. The above-acquired rBSA-FA was dissolved in the PBS solution and then mixed with GO-AuNSs for 8 h. Then, the mixture was centrifuged and washed with PBS at least three times to obtain the GO-AuNSs@rBSA-FA nanocomposites. Finally, the synthesized nanomaterials were stored in a 4 °C refrigerator away from light.

### 2.4. Assembly of Electrochemical Cytosensors Based on GO-AuNSs@rBSA-FA Nanocomposites

The glassy carbon electrode (GCE) was polished with aluminum powder and washed with alcohol and deionized water under ultra-sonication. Then, a gold nano-layer was electroplated on the surface of the GCE according to our previous work [39]. Briefly, under a three-electrode system, the polished glassy carbon electrode was immerged in 1% HAuCl_4_ that acted as the electrolyte to generate a Au layer on the surface of the working electrode. The Au layer-modified working electrode was obtained by electrodepositing at a fixed potential −0.2 V in i-t mode for 300 s (abbreviated as DpAu). Following the plating step, GO-AuNSs@rBSA-FA (2 mg/mL, 3 μL) was dropped on the surface of the coating and incubated for 3 h under room temperature. The excess GO-AuNSs@rBSA-FA nanocomposites on the surface of the Au electrode were washed off by fresh PBS solution to obtain the GCE/DpAu/GO-AuNSs@rBSA-FA sensing interface Subsequently, to block nonspecific binding sites of the electrode, 5 µL of BSAT solution (0.5% BSA in 1% Tween-20) was used to modify the electrode 1 h (abbreviated as GCE/DpAu/GO-AuNSs@rBSA-FA/BSAT). Finally, the above-obtained electrodes were incubated with the cell solution at 37 °C for 1 h. The electrochemical tumor cell sensors were finally constituted by an electrochemical workstation in which DpAu/GO-AuNSs@rBSA-FA acted as the working electrode, platinum wire acted as the auxiliary electrode, and the Ag/AgCl electrode was used as the reference electrode.

### 2.5. Electrochemical Detection of MGC-803 Cells

MGC-803 cells were derived from gastric cancer patients and have been used in scientific research for a long time in our laboratory. MGC-803 cells were cultured in a medium with 10% fetal bovine serum, 100 U mL^−1^ penicillin, 0.1 mg mL^−1^ streptococcus, and high amounts of sugar, which were placed in a cell incubator (temperature: 37 °C, carbon dioxide concentration: 5%). After the cells overgrew in the culture dish, they were digested with trypsin to further realize passage or cryopreservation. 

The Fe(CN)_6_^3−^ (5 mM) containing KCl (0.1 M) (pH 7.4) was used to obtain a cyclic voltammetry (CV) solution for the electrochemical CV curve and differential pulse voltammetry (DPV) curve measurement. The AC solution supporting electrochemical impedance curve measurement was obtained from a 5 mM K_3_Fe(CN)_6_/K_4_Fe(CN)_6_ (1:1) and 0.1 M KCl solution (pH 7.4). The cultured MGC-803 cells were collected by trypsin and diluted with PBS solution to form cell suspensions in various concentrations; 5 μL of cell suspension was dropped on the GCE/DpAu/GO-AuNSs@rBSA-FA/BSAT electrode and then placed at 37 °C for 1-h incubation. After the excess cells were washed off from the electrode surface, electrochemical detection experiments could be carried out on the sensor platform.

## 3. Results and Discussion

### 3.1. Characterization of GO-AuNSs@rBSA-FA

The morphology of the GO nanosheet, GO-AuSeeds, and GO-AuNSs was characterized by TEM and SEM (Figure 2a–d). As shown in Figure 2a, the silk-like morphology of GO indicated its large surface, which would provide sufficient growth sites for gold seeds and AuNSs. After adding HAuCl_4_ to the GO solution, numerous spherical gold nanoparticles were generated on the surface of GO and formed GO-AuSeeds (Figure 2b). Furthermore, according to the seed-mediated method, the larger gold nanomaterials on the GO surface then evolved into a star shape (Figure 2c), which illustrated the successful growth of a large number of AuNSs and uniform size of GO-AuNSs. Furthermore, the morphology and structure of GO-AuNSs were explored by SEM (Figure 2d). As shown in the SEM picture, massive AuNSs on the GO surface were observed, further indicating the feasibility of our experimental synthesis. The EDX energy spectrum of GO-AuNSs is shown in Figure 2e, indicating that the GO-AuNS nanocomposites were composed of elements Au, C, and O, in which the proportion of element Au was the highest, indicating large quantities of AuNSs on the surface of GO. In this nanocomposite, the silk-like GO and the AuNSs could greatly enhance the conductivity of the final cytosensors. Furthermore, the GO could also provide various functional groups for further modification.

The UV–vis absorption characteristics of GO, GO-AuSeeds, GO-AuNSs, rBSA, FA, rBSA-FA, and GO-AuNSs@rBSA-FA were studied as well (Figure 3). According to Figure 3a, the UV–vis spectrum of rBSA exhibited a distinct peak centered at 279 nm (curve A in Figure 3a), and FA displayed an ultraviolet absorption peak at 280 nm and 350 nm (curve B) [38]. As the FA was cross-linked onto rBSA, the generated rBSA-FA complex exhibited an additional characteristic peak at 350 nm compared with bare rBSA, indicating the successful formation of rBSA-FA [40]. 

Moreover, as demonstrated in Figure 3b, GO showed a typical absorption peak at 239 nm (curve D in Figure 3b) [41]. As the Au seeds were directly formed on the surface of GO via sonication in an alkaline environment without adding traditional chemical reductants and surfactants, the UV–vis spectrum displayed obvious peaks at 247 nm and 520 nm corresponding to GO and AuNPs, respectively (curve E). When the AuNSs were finally generated on GO according to the seed-mediated method, the ultraviolet absorption peak moved from 534 nm to 800 nm and showed a much broader absorption owing to specific localized surface plasmon resonance (LSPR) of Au nanomaterials in a star-like shape (curve F) [42,43]. Finally, as the above-obtained rBSA-FA was modified onto GO-AuNSs via Au–S bonds, there were specific UV peaks around 260 nm and 800 nm due to the existence of rBSA, FA, and AuNSs, proving the successful construction of the final nanocomposite GO-AuNSs@rBSA-FA.

### 3.2. Electrochemical Performances of GO-AuNSs@rBSA-FA

In this protocol, electrochemical impedance spectroscopy (EIS) and CV were carried out to investigate the surface characteristic and the electron transfer behaviors of the electrode surface. The EIS was measured in 5 mM K_3_Fe(CN)_6_/K_4_Fe(CN)_6_ (1:1) and 0.1 M KCl solution (pH 7.4) to reflect the electron-transfer resistance change of every fabrication step, while the CV analysis was performed in Fe(CN)_6_^3−^ (5 mM) containing KCl (0.1 M) (pH 7.4). The equivalent circuit in Figure 4b was composed of double layer capacitance (Cdl), diffusion impedance (Zw), polarisation resistance (Rp), and electron-transfer resistance (Rct). In the circuit, Cdl represented the capacitance between the working electrode and electrolyte, Zw referred to the impedance of reactant diffusion from the solution body to the electrode reaction interface, and Rp was the resistance to electrode reactions induced by electrochemical polarisation and diffusion control. Rct reflected the ease of charge transfer across the interface between the electrode and the electrolyte solution during the electrode process. The Nyquist diagrams of the EIS results are displayed in Figure 4a, and circular diameters in the EIS curve reflect the charge transfer resistance of the electrodes (R_ct_). As shown in Figure 4a, the resistance of the bare GCE (R_et_ = 340.2 Ω) was obviously higher than that of gold-plated GCE (R_ct_ = 91.3 Ω), indicating the favorable conductivity of the gold layer. After modified with GO-AuNSs@rBSA-FA nanocomposites, a marked increase in the R_et_ value of the DpAu/GO-AuNSs@rBSA-FA electrode (R_ct_ = 717.6 Ω) was obtained compared with gold-plated GCE, which might be related to the activity of rBSA-FA to hinder electron transfer. To block the nonspecific sites of electrodes, the BSAT was finally assembled on the surface of the electrode, and the R_ct_ of the DpAu/GO-AuNSs@rBSA-FA/BSAT electrode reached 1091 Ω, ascribed to the electronically inert character of BSA. As the gastric cancer cells were captured by our assembled electrochemical sensors due to specific immunoreaction between FA and antigen, the R_ct_ was increased to 1293 Ω, indicating that the cells, biological macromolecules, could seriously hinder the electron transfer tunnel. The capture ability of GO-AuNSs@rBSA-FA to MGC-803 cells was revealed, which further illustrated the feasibility of the experimental design.

Moreover, as shown in Figure 4b, the CV curve was also used to reflect the current change of each modified procedure. The peak current of the bare GCE was 97.53 μA, while the Au layer-modified electrode was 131.1 μA, showing that the conductivity of the Au layer was superior to that of the bare GCE, and the smaller current (56.72 μA) of the GO-AuNSs@rBSA-FA electrode indicated the successful immobilization of the nanocomposites. As the BSAT was attached to the electrode surface, the current of the electrode was reduced to 48.69 μA, proving the successful fabrication of BSAT. A lower current of 39.7 μA of the electrode was acquired after incubation with gastric cancer cells, proving that the electrochemical cytosensors based on GO-AuNSs@rBSA-FA in this work could be applied for MGC-803 cell capture and detection.

### 3.3. Electrochemical Detection of Gastric Cancer Cells

In this work, the electrochemical differential pulse volt (DPV) method was employed to detect gastric cancer cells (MGC-803). Briefly, 5 μL of fresh cell PBS solution in various concentrations was separately added to DpAu/GO-AuNSs@rBSA-FA/BSAT-modified electrodes and then incubated for at least 1 h. The DPV measurement results of various electrodes for MGC-803 cell determination are shown in Figure 5, and the corresponding calibration curve between the DPV current value and cell concentrations is illustrated in Figure 5a. The DPV current value was inversely proportional to the MGC-803 cell concentration due to the poor conductivity of cancer cells, and the DPV current value constantly decreased with the increase in the cell concentration when the gastric cancer cells were effectively captured by the proposed sensors (the current value list in Table 1). All above-mentioned results indicated that our designed electrochemical sensors based on GO-AuNSs@rBSA-FA could be used to detect MGC-803 cells. Moreover, the linear relationship between the cell concentration and DPV current value was further explored. As shown in Figure 5b, the DPV curve value was linearly associated with the logarithm of the MGC-803 cell concentrations (C_MGC-803_) in the range from 3 × 10^2^ to 7 × 10^6^ cell/mL, and the determination of this fabricated cytosensor was 1 × 10^2^ cell/mL. The linear regression equation is *I* = 7.4983 × Log(C_MGC-803_) − 87.26046 (correlation coefficient R^2^ = 0.97305). Compared with other reported electrochemical assays (Table 2), this work had a low determination limit and wide analysis range, indicating the good electrochemical properties of our electrochemical cytosensor for MGC803 cell quantitative determination. Furthermore, in situ growth of gold seeds on GO effectively enhanced the binding force between the AuNSs and GO rather than relying on electrostatic adsorption alone. Therefore, the following modification of FA would be more reliable. The GO-AuNSs@rBSA-FA increased the specific surface area, biocompatibility, and specificity of the cytosensor, thus providing a reference significance for clinical detection of cancer cells.

The stability and specificity of the nanoplatform were analyzed as well. The GO-AuNSs@rBSA-FA electrode was stored in a 4 °C refrigerator and measured with the CV method every three days. The current value measured for the first time (day 0) was labeled as 100%. As shown in Figure 6a, the current measured on days 3, 6, 9, 12, and 15 reached 97.67%, 94.66%, 90.2%, 88.88%, and 87.29%, respectively, compared with day 1, demonstrating the considerable stability of our electrochemical cytosensors. In this work, the GO provided a strong load structure for the growth of AuNSs, thus hindering the gathering of the scattered AuNPs and ensuring the stability of the material structure. Moreover, the modification of rBSA-FA further enhanced the stability of the nanomaterials based on the Au-S bond. To study the specificity of the nanocomposites, the GO-AuNSs@rBSA-FA electrode was employed to capture different kinds of cells, and the resistance increase of cell-modified electrodes relative to the DpAu/GO-AuNSs@rBSA-FA/BSAT electrodes was measured by the EIS method (Figure 6b). The cytosensors were modified with human lung cancer cells (A549), human gastric mucosa cell (GES1), MGC803 cells, and a cell mixture, and the relative increases of these electrodes were 2.87%, 3.63%, 19.27%, and 17.13%, respectively (relative increase rate: EIS of cytosensor modified with cells relative to without cells). Since the amount of FR was negligible in A549 and GES1, the change in the resistance of the cell-modified working electrode was much smaller compared with that modified with MGC803 cells. The results have testified that our constructed cytosensors have an excellent specificity for MGC803 cell detection.

## 4. Conclusions

In conclusion, we fabricated a GO-AuNSs@rBSA-FA nanomaterial-based electrochemical cytosensor for MGC-803 cancer cell analysis. The nanocomposites had a large surface area, good biocompatibility, and excellent stability. In addition, the nanoplatform could specifically capture the cancer cells based on the FA-FR binding. The cytosensor has a quantitative detection of cells in a wide linear range of 3 × 10^2^~7 × 10^6^ cell/mL with a limit of 1 × 10^2^ cell/mL (S/N = 3). In this work, our electrochemical sensor based on GO-AuNSs@rBSA-FA showed favorable stability, outstanding specificity, and good electrochemical characteristics for gastric cancer cell MGC-803 analysis, proving a promising detection strategy for cancer cells. Furthermore, it provided a potential strategy for early cancer diagnosis, as well as the possibility for clinical detection in the future.

## Figures and Tables

**Figure 1 sensors-22-02783-f001:**
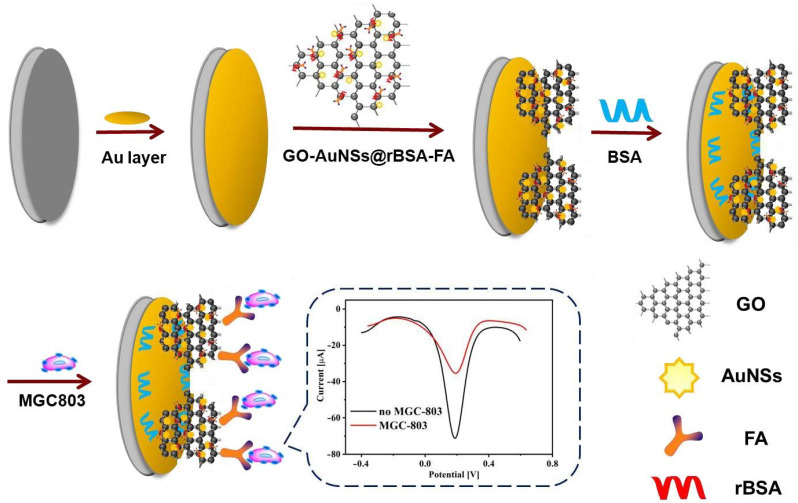
Schematic illustrating the fabrication procedure of the electrochemical cytosensor based on GO-AuNSs@rBSA-FA for MGC-803 cell detection.

**Figure 2 sensors-22-02783-f002:**
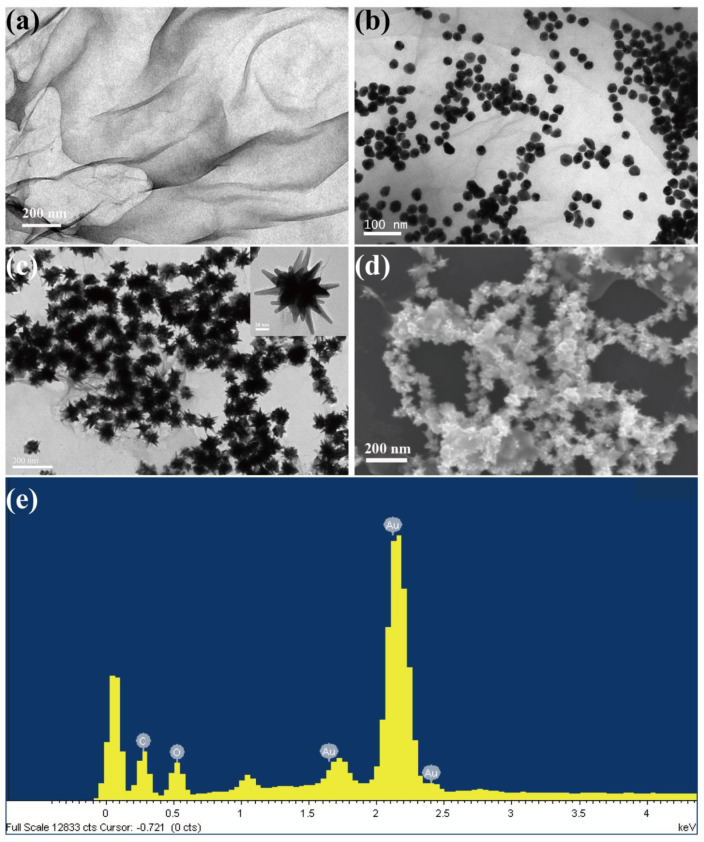
TEM image of (**a**) GO, (**b**) GO-AuSeeds and (**c**) GO-AuNSs. SEM image (**d**), and EDX spectrum (**e**) of GO-AuNSs.

**Figure 3 sensors-22-02783-f003:**
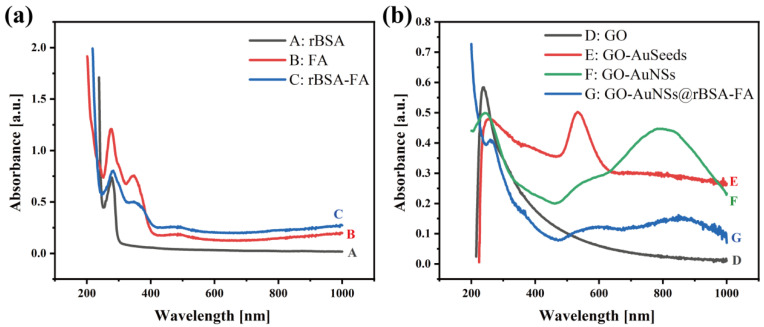
UV–vis absorption spectra of (**a**) GO, GO-AuSeeds, GO-AuNSs, GO-AuNSs@rBSA-FA and (**b**) rBSA, FA, rBSA-FA.

**Figure 4 sensors-22-02783-f004:**
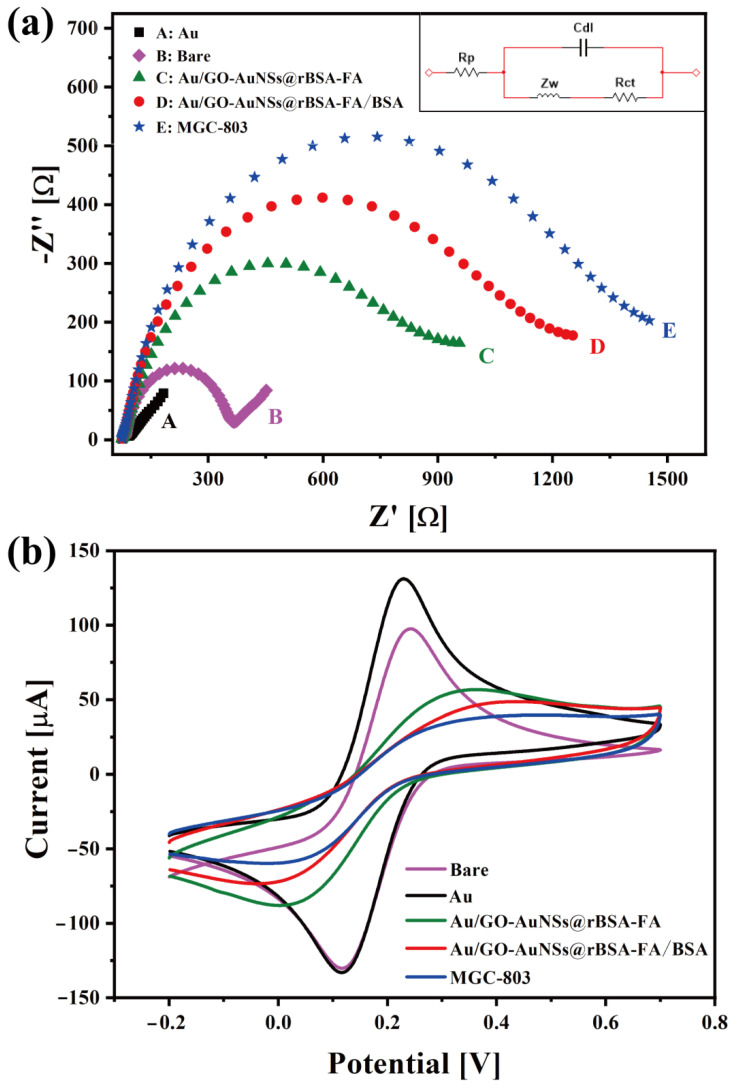
The (**a**) EIS graphs with an equivalent circuit inset in pictures a and (**b**) CV curves of different electrodes in various assembly steps. Among them, purple curves represent GCEs; black curves represent Au electrodes; green curves represent DpAu/GO-AuNSs@rBSA-FA electrodes; red curves represent DpAu/GO-AuNSs@rBSA-FA/BSAT electrodes; and blue curves represent DpAu/GO-AuNSs@rBSA-FA/BSAT electrodes modified with MGC-803 cells.

**Figure 5 sensors-22-02783-f005:**
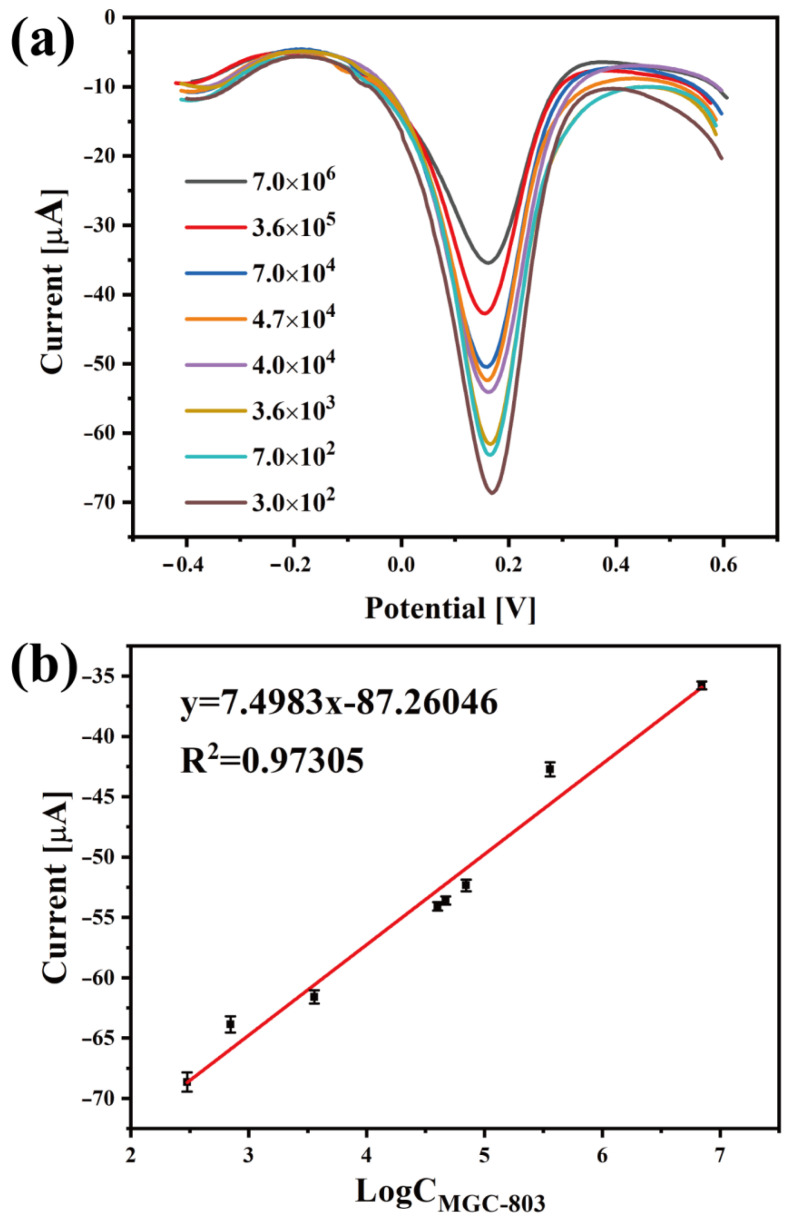
(**a**) The DPV currents of GO-AuNSs@rBSA-FA-labeled electrodes for detecting different concentrations of MGC-803 cells in the range from 3 × 10^2^–7 × 10^6^ cell/mL. (**b**) The calibration curve of DpAu/GO-AuNSs@rBSA-FA/BSAT electrodes with respect to MGC-803 cells.

**Figure 6 sensors-22-02783-f006:**
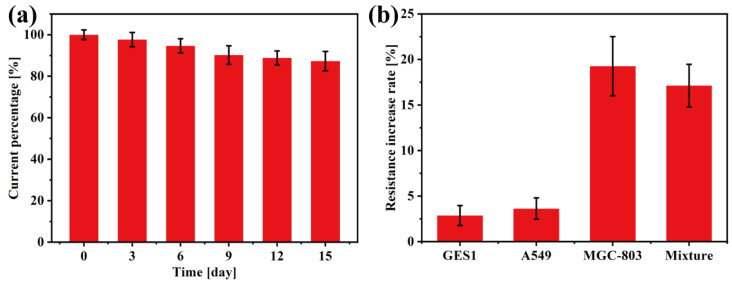
(**a**) The change in the current value of GO-AuNSs@rBSA-FA-labeled electrodes with time. (**b**) The resistance increase in GO-AuNSs@rBSA-FA-labeled electrodes modified with different cells and their mixtures.

**Table 1 sensors-22-02783-t001:** The DPV currents of GO-AuNSs@rBSA-FA-labeled electrodes for various concentrations of MGC-803 cells.

Concentration of MGC-803 Cells (Cell/mL)	3 × 10^2^	7 × 10^2^	3.6 × 10^3^	4 × 10^4^	4.7 × 10^4^	7 × 10^4^	3.6 × 10^5^	7 × 10^6^
The maximum value (μA)	68.65	63.87	61.59	54.09	53.6	52.35	42.72	35.78

**Table 2 sensors-22-02783-t002:** The performance of this cytosensor compared with others reported.

Detection Technique	Nanomaterials	Linear Range (Cell/mL)	Detection Limit (Cell/mL)	References
DPV	Peptides–single-walled carbon nanotubes	1.0 × 10^3^~1.0 × 10^7^	620	[44]
EIS, CV	AuNPs–chitosan	1.34 × 10^4^~1.34 × 10^8^	871	[45]
EIS	CNTs@PDA-FA	5 × 10^2^~5 × 10^6^	500	[46]
ASV	SiO_2_@QDs–ConA	1 × 10^3^~1 × 10^7^	1000	[47]
ECL	Ru–DNA–AuNPs	5 × 10^2^~1 × 10^5^	300	[48]
Colorimetric	GO-AuNCs	1 × 10^3^~2 × 10^5^	-	[49]
EIS	FA/PEI/CMC-G	5 × 10^2^~5 × 10^6^	500	[50]
EIS	Hyaluronate/graphene	5 × 10^2^~5 × 10^6^	100	[51]
DPV	GO-AuNSs@ rBSA-FA	3 × 10^2^~7 × 10^6^	100	This work

CNTs, carbon nanotubes; PDA, polydopamine; ASV, anodic stripping voltammetry; QDs, quantum dots; PEI, polyethyleneimine; CMC-G, carboxymethyl chitosan-functionalized graphene; ConA, concanavalin A; EIS, electrochemical impedance spectra.

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
