# Peer review of "Electrochemical Cytosensor Based on a Gold Nanostar-Decorated Graphene Oxide Platform for Gastric Cancer Cell Detection"

_sensors, 2022, doi:10.3390/s22072783_

Round 1

Reviewer 1 Report

The manuscript tilted "Electrochemical Cytosensor Based on Gold Nanostars-Deco-2 rated Graphene Oxide Platform for Gastric Cancer Cell Detec-3 tion" by Zhang et al proposed effectively and sensitively detecting cancer cell by GO-AuNSs nanocomposite. The sensor working condition and performances were evaluated. In addition, It provides sufficient contents. Therefore, it is considered that this study has sufficient qualifications as a thesis with only a few minor supplements. It could be published after address the following comments:

  1. I would like to describe why significant value in the LOD 100 cell/mL of MGC-803 cell had in cancer incidence.
  2. In fig. 3, it seems better to attach a reference related to each peak rather than describing the location of each UV peak in the synthesis process of each material.
  3. Table 2 and graph of 5 shows different the linear range.
  4. In Table 2, it would be good to add the paper of cytosensor using graphene oxide (GO) and reduced graphene oxide(RGO). It shows the excellence of this study.
  5. No. 34 reference error? The overall reference format needs to be unified.

Author Response

Responses to reviewer's comments (Review ID: sensors-1633995)

Dear reviewer,

Thank you for your rapid response to our manuscript entitled “Electrochemical Cytosensor Based on Gold Nanostars-Decorated Graphene Oxide Platform for Gastric Cancer Cell Detection (Manuscript ID: sensors-1633995). We are particularly grateful to you for your detailed and constructive comments, which are all valuable and very helpful for revising our paper and the important guiding significance to our researches. We have studied revision comments carefully and tried to address all your concerns. The relevant changes have been marked with red word. The detailed revisions and explanations to each point raised by you are demonstrated as following:

To: Reviewer #1’s Comments

1. I would like to describe why significant value in the LOD 100 cell/mL of MGC-803 cell had in cancer incidence.

Answer 1: Thank you for your constructive comment for this manuscript. In this paper, Gold Nanostars-Decorated Graphene Oxide Platform was innovatively synthesized to construct electrochemical sensor, which is expected to be able to capture gastric cancer cells with favorable specificity and sensitivity. In conventional studies, the electrode is immersed into the cell suspension to absorb cells with a long time.While in this paper, to quickly analyze the analytical performance of the sensor, we modified the cells layer by dropping method, in which 5 μL of cell suspension was dropped onto GCE/DpAu/GO-AuNSs@rBSA-FA/BSAT electrode. And the LOD 100 cell/mL of MGC-803 cell means that only 0.5 cell per 5 μL is attached to the electrode surface. In future study, fully infiltrated modification methods of cells could be used during sensors’ construction.  

2. In fig. 3, it seems better to attach a reference related to each peak rather than describing the location of each UV peak in the synthesis process of each material.

Answer 2: We sincerely appreciate the valuable comments of reviewer. We have attached references related to the UV peak of FA (38), rBSA-FA(40), GO(41) and AuNSs (42,43), and present description of each peak in detail.

3. Table 2 and graph of 5 shows different the linear range.

Answer 3: Thank you for your question. The linear range in Table 2 is 3×102~7×106 cell/mL, which was the concentration range of MGC-803 cells (CMGC-803). The x-axis in Fig 5 is LogCMGC-803, and the linear range in this figure was Log 3×102~ Log 7×106. We have corrected the unit symbols of x-axis in Fig 5.

4. In Table 2, it would be good to add the paper of cytosensor using graphene oxide (GO) and reduced graphene oxide (RGO). It shows the excellence of this study.

Answer 4: Thank you for your comments. Some papers about cytosensors by using graphene have been inserted into table 2.

5. No. 34 reference error? The overall reference format needs to be unified.

Answer 5: Sorry for our careless. We have replenished the reference (39) here.

We hope this manuscript has improved satisfactorily. Thank you very much for your effort in evaluation of our manuscript.

Sincerely yours,

Daxiang Cui

Reviewer 2 Report

The manuscript describes the fabrication, characterization and use of an glassy carbon electrode modified with GO and AuNS to capture tumors cells. The capture is based on te presence of folic acid receptors on these cells. By decorating the nanoobjects with folic acid, the cells are bound to the electrode surface, blocking it as shown by EIS and CV measurements.

In it current form, the manuscript lacks critical information. 

My main concern is that the authors do not provide any evidence of the benefit of their electrode architecture over the state of the art. It is therefore impossible to evaluate its performances in comparison to the other technical solutions or nano-objects, which is important to justify the advancement of the field.

This point is critical as Figure 4 shows that the proposed assembly dramatically hinders the electrode response. As the sensor is based on partial blocking of the electrode by cell capture, this low electrochemical activity reduces the sensitivity of the sensor. As we cannot compare this response to the one of other electrode structures, it is impossible to evaluate the performace of the sensor in the context of the state-of-the-art.

As a consequence, the limit of detection is 100 cells per ml, which is mostly to high for circulating tumor cells.

On figure 5a, you claim you are presenting DPV, however the axes are labeled absorbamce and wavelength.

Author Response

Responses to reviewers comments (Review ID: sensors-1633995)

Dear reviewer,

Thank you for your rapid response to our manuscript entitled “Electrochemical Cytosensor Based on Gold Nanostars-Decorated Graphene Oxide Platform for Gastric Cancer Cell Detection (Manuscript ID: sensors-1633995). We are particularly grateful to you for your detailed and constructive comments, which are all valuable and very helpful for revising our paper and the important guiding significance to our researches. We have studied revision comments carefully and tried to address all your concerns. The relevant changes have been marked with red word. The detailed revisions and explanations to each point raised by you are demonstrated as following:
To: Reviewer #2’s Comments

The manuscript describes the fabrication, characterization and use of an glassy carbon electrode modified with GO and AuNS to capture tumors cells. The capture is based on the presence of folic acid receptors on these cells. By decorating the nanoobjects with folic acid, the cells are bound to the electrode surface, blocking it as shown by EIS and CV measurements. In it current form, the manuscript lacks critical information.  My main concern is that the authors do not provide any evidence of the benefit of their electrode architecture over the state of the art. It is therefore impossible to evaluate its performances in comparison to the other technical solutions or nano-objects, which is important to justify the advancement of the field.

1. This point is critical as Figure 4 shows that the proposed assembly dramatically hinders the electrode response. As the sensor is based on partial blocking of the electrode by cell capture, this low electrochemical activity reduces the sensitivity of the sensor. As we cannot compare this response to the one of other electrode structures, it is impossible to evaluate the performace of the sensor in the context of the state-of-the-art.

Answer 1: Thank you for your favorable comments for our work. The commonly used approaches to improve the sensitivity of sensors might include two kinds of strategies: increasing the conductivity of the sensors and increasing the specific surface area of the sensors. In this paper, owing to the exist of non-conductive biomolecules (BSA and FA), the conductivity of the electrochemical sensors have been affected. However, the prepared Gold Nanostars-Decorated Graphene Oxide platform also increased the specific surface area of final sensors. Moreover, the used nanocomplex would further enhance the biocompatibility and specificity due to the exist of BSA and FA in respective. Without using the Gold Nanostars-Decorated Graphene Oxide platform, FA and BSA were also needed for sensors’ construction, which might arouse larger impedance.

2. As a consequence, the limit of detection is 100 cells per ml, which is mostly toohigh for circulating tumor cells.

Answer 2: Thank you for your constructive comment for this manuscript. In this paper, Gold Nanostars-Decorated Graphene Oxide Platform was innovatively synthesized to construct electrochemical sensor, which is expected to be able to capture gastric cancer cells with favorable specificity and sensitivity. In conventional studies, the electrode is immersed into the cell suspension to absorb cells with a long time.While in this paper, to quickly analyze the analytical performance of the sensor, we modified the cells layer by dropping method, in which 5 μL of cell suspension was dropped onto GCE/DpAu/GO-AuNSs@rBSA-FA/BSAT electrode. And the LOD 100 cell/mL of MGC-803 cell means that only 0.5 cell per 5 μL is attached to the electrode surface. In future study, fully infiltrated modification methods of cells could be used during sensors’ construction.  

3. On figure 5a, you claim you are presenting DPV, however the axes are labeled absorbance and wavelength

Answer 3: Sorry for our careless. We have corrected the y-axis into ‘Current’ and x-axis into ‘Potential’ in Fig 5a.

We hope this manuscript has improved satisfactorily. Thank you very much for your effort in evaluation of our manuscript.

Sincerely yours,

Daxiang Cui

Reviewer 3 Report

The work title “Electrochemical Cytosensor Based on Gold Nanostars-Decorated Graphene Oxide Platform for Gastric Cancer Cell Detection” is a great work that demonstrated the construction of a novel platform for electrochemical detections of gastric cancer cell. The material is well characterized and the application is well described. For these reasons, I suggest to accept the work in Sensors.

There are just some questions for the authors:

  1. In abstract section (line19) change the term “paltform” for “platform”.
  2. DPV is not a technology. DPV is a technique that measure the obtained current of redox process. The term ‘technology’ must be removed.
  3. In line 52-53 the authors have been said that GO has abundant functional groups and high hydrophobicity. It’s not real, as GO has many -COOH groups, the whole structure have been more hydrophilic. Verify this information.
  4. In the first paragraph, the authors need to clarify the choice of FA. Verify some references thar could help and cite them: i) 2116/analsci.20P297; ii) 10.1016/j.talanta.2022.123278; iii) 10.1016/j.jfca.2020.103511
  5. In Fig 4, the authors show the impedance data. It would be interesting to show the equivalent circuit inset of Fig4B and explain all the circuit components.
  6. In Fig 5, I think the authors say y-axis is Absorbance (instead Current) and x-axis as wavelength (instead Potential vs. (reference electrode)). Correct it.
  7. What is the advantage of this electrode compared to the ones observed in the table 2. Focuses in the modification of material and facilities, not just in the detection limit.

Author Response

Responses to reviewers comments (Review ID: sensors-1633995)

Dear reviewer,

Thank you for your rapid response to our manuscript entitled “Electrochemical Cytosensor Based on Gold Nanostars-Decorated Graphene Oxide Platform for Gastric Cancer Cell Detection (Manuscript ID: sensors-1633995). We are particularly grateful to you for your detailed and constructive comments, which are all valuable and very helpful for revising our paper and the important guiding significance to our researches. We have studied revision comments carefully and tried to address all your concerns. The relevant changes have been marked with red word. The detailed revisions and explanations to each point raised by you are demonstrated as following:
To: Reviewer #3’s Comments

1. In abstract section (line19) change the term “paltform” for “platform”.

Answer 1: Sorry for our careless. We have corrected the spelling error, and change “paltform” to “platform”.

2. DPV is not a technology. DPV is a technique that measure the obtained current of redox process. The term ‘technology’ must be removed.

Answer 2: Thank you for your good comments. We have removed the term ‘technology’.

3. In line 52-53 the authors have been said that GO has abundant functional groups and high hydrophobicity. It’s not real, as GO has many -COOH groups, the whole structure have been more hydrophilic. Verify this information.

Answer 3: Thank you for your professional suggestion. It is right that GO has many -COOH groups, the whole structure have been more hydrophilic. Sorry for our careless, and we have corrected ‘hydrophobicity’ into ‘hydrophilicity’.

4. In the first paragraph, the authors need to clarify the choice of FA. Verify some references thar could help and cite them: i) 2116/analsci.20P297; ii) 10.1016/j.talanta.2022.123278; iii) 10.1016/j.jfca.2020.103511

Answer 4: Thank you for your favorable suggestion. We clarified the choice of FA at the end of first paragraph, Whats more, the design of folic acid (FA) modified……in recent years. The description of FA was also improved according to theses references in the third paragraph and these references were cited (31-33).

5. In Fig 4, the authors show the impedance data. It would be interesting to show the equivalent circuit inset of Fig4B and explain all the circuit components.

Answer 5: We appreciated the valuable suggestions and we have inseted the equivalent circuit into Fig4B and given the explanation of all the circuit components before the analysis of Fig.4a,The equivalent circuit in Fig.4b was composed of double layer capacitance (Cdl) …… between the electrode and the electrolyte solution during the electrode process’.

6. In Fig 5, I think the authors say y-axis is Absorbance (instead Current) and x-axis as wavelength (instead Potential vs. (reference electrode)). Correct it.

Answer 6: Sorry for our careless. We have corrected the y-axis into ‘Current’ and x-axis into ‘Potential’ in Fig 5a.

7. What is the advantage of this electrode compared to the ones observed in the table 2. Focuses in the modification of material and facilities, not just in the detection limit.

Answer 7: We sincerely appreciate the valuable comments. We have supplemented the advantages of the modified material following the comparison of detection limit, ‘Whats more, in situ growth of gold seeds …… biocompatibility and specificity of the cytosensor’.

We hope this manuscript has improved satisfactorily. Thank you very much for your effort in evaluation of our manuscript.

Sincerely yours,

Daxiang Cui
